# Heat Stroke in the Work Environment: Case Report of an Underestimated Phenomenon

**DOI:** 10.3390/ijerph20054028

**Published:** 2023-02-24

**Authors:** Maricla Marrone, Luigi Buongiorno, Pierluigi Caricato, Fortunato Pititto, Benedetta Pia De Luca, Carlo Angeletti, Gabriele Sebastiani, Eliano Cascardi, Giuseppe Ingravallo, Alessandra Stellacci, Gerardo Cazzato

**Affiliations:** 1Section of Legal Medicine, Department of Interdisciplinary Medicine, University of Bari “Aldo Moro”, 70124 Bari, Italy; 2Department of Medical Sciences, University of Turin, 10124 Turin, Italy; 3Pathology Unit, FPO-IRCCS Candiolo Cancer Institute, 10060 Candiolo, Italy; 4Section of Molecular Pathology, Department of Precision and Regenerative Medicine and Ionian Area (DiMePRe-J), University of Bari “Aldo Moro”, 70124 Bari, Italy

**Keywords:** heat-related injuries, work-related deaths, climate change, autopsy, forensic pathology

## Abstract

Average global temperatures continue to trend upward, and this phenomenon is part of the more complex climate change taking place on our planet over the past century. Human health is directly affected by environmental conditions, not only because of communicable diseases that are clearly affected by climate, but also because of the relationship between rising temperatures and increased morbidity for psychiatric diseases. As global temperatures and the number of extreme days increase, so does the risk associated with all those acute illnesses related to these factors. For example, there is a correlation between out-of-hospital cardiac arrest and heat. Then, there are pathologies that recognize excessive heat as the main etiological agent. This is the case with so-called “heat stroke”, a form of hyperthermia accompanied by a systemic inflammatory response, which causes multi-organ dysfunction and sometimes death. Starting with a case that came to their attention of a young man in good general health who died while working unloading fruit crates from a truck, the authors wanted to express some thoughts on the need to adapt the world of work, including work-specific hazards, in order to protect the worker exposed to this “new risk” and develop multidisciplinary adaptation strategies that incorporate climatology, indoor/building environments, energy use, regulatory perfection of work and human thermal comfort.

## 1. Introduction

Average global temperatures continue to trend upward. This is part of the more complex phenomenon of climate change taking place in the last century on our planet. Globally, compared with the pre-industrial era (1850–1900), the global temperature is 1.2 °C higher. The past 7 years have been the hottest 7 years on record, and so far, 2020 has been the hottest year ever [1]. Italian data agree with international data, with the 2020 temperature in the regional capitals increasing by +1.2 °C from the average of 1971–2000 [2]. One of the most widely accepted hypotheses to explain the reasons for global warming concerns the increase in anthropogenic emissions of greenhouse gases, such as carbon dioxide, methane and nitrous oxide. Greenhouse gases increase the average temperature by trapping more heat in the lower atmosphere. If this trend is not halted, climate change will lead to increasingly frequent extreme weather events (floods and heat waves) [3]. Human health is directly affected by environmental conditions. Climate change alters the environment, implying variations in agricultural yields and, consequently, periods of famine and malnutrition [4]. Communicable diseases are also affected by climate. These include vector-borne diseases such as leptospirosis, of which there are more cases after a flood, or diseases transmitted by ticks and mosquitoes, which benefit from a warmer climate for their life cycle and expansion of their geographic distribution [5]. Other diseases such as typhoid, cholera, malaria, dengue and West Nile virus infection may also increase in prevalence as a result of climate change [6]. Significant associations also emerge between increased temperature and increased morbidity for psychiatric conditions, such as mood disorders, organic mental disorders, schizophrenia, neurotic and anxiety disorders [7].

As global temperatures and the number of extreme heat days increase, so does the risk associated with all those acute illnesses related to these factors. For example, there is a correlation between out-of-hospital cardiac arrest and heat. This association is most significant in diabetic populations and those with cardiac comorbidities [8]. A 1 °C increase in temperature has also been associated with a significant increase in morbidity due to arrhythmias, cardiac arrest and coronary artery disease [9].

Some diseases precisely recognize excessive heat as the main etiological agent. This is the case with so-called “heat stroke”, a form of hyperthermia accompanied by a systemic inflammatory response, resulting in multi-organ dysfunction and sometimes death. Symptoms include hyperthermia (body temperature > 40 °C) and altered mental status; diagnosis is clinical. Treatment includes rapid external cooling, resuscitation with EV fluids and supportive therapy as required in organ failure [10].

In general, there is a correlation between climatic parameters (e.g., maximum temperature and humidity index) and mortality [11], with an increased risk of mortality increased between 1% and 3% per 1 °C increase in temperature [12]. In the occupational setting, several studies have correlated excessive heat with increased occupational injuries [13,14], with increased risk especially among construction workers [15].

In this study, we present the case of a person who died in a work environment due to heat stroke, opening insights and reflections on the topic.

## 2. Case Presentation

This case involves a male individual, age 38, with no chronic medical conditions and in good general health who, in the early afternoon of 25 August 2020, while working unloading fruit crates from a truck, suddenly died.

The subject performed his duties at a private company of Italy, from which he had been employed about 15 years earlier. The work activity consisted of driving the vehicle (truck driver) and transporting the goods. He was, therefore, a transporter truck driver. He provided, in addition, the unloading of goods, and used to perform this work activity alone.

The survey was conducted in the countryside of Italy. We arrived at the site about 90 min after the death (about 4:30 p.m.). Some passers-by reported that the illness happened about 90 min before our arrival. The rectal temperature was 38.5 °C, and cadaveric rigidity and hypostases were absent. On the occasion, we found high ambient temperature (39.1 °C). High air humidity was also present. In fact, the hygrometer provided measured 80% humidity in the air. Ventilation was almost absent. The thanatochronological parameters were compatible with the time data referred by the witnesses. The body was wearing a gray cotton T-shirt, a pair of dark blue jeans and shoes of a dark-colored leather-like material. Nothing significant was found at the external cadaveric inspection (Figure 1, Figure 2 and Figure 3).

At autopsy, a condition of diffuse organic congestion was found, particularly pronounced in the brain, lung, liver, kidney and gastric areas. The heart also appeared flaccid and exhausted (Figure 4 and Figure 5), a fact that contrasted with the patient’s young age and absence of known pathology. Finally, the kidneys also showed altered normal architecture. In this view, there were no morphological alterations to suppose other causes of a different nature, except for an important and widespread presence of vascular ectasia dilatation phenomena related to blood stasis (Figure 6 and Figure 7), and consequent tissue congestion and imbibition.

Furthermore, no signs of trauma or other pathology were detected to justify the rapid death. Reading the histological preparations allowed confirmation of widespread edema and stasis, indicative of cardiovascular collapse on a functional basis (Figure 8 and Figure 9). In relation to the macroscopic data (diffuse congestion, absence of trauma and absence of gross acute pathology capable of determining rapid death), histological data (stasis, diffuse edema) and reported circumstantial elements that depicted the worker’s significant physical exertion, together with the extreme climatic conditions due to the intense heat of the day, it was possible to determine the cause of death as acute cardio-circulatory arrest on a functional basis.

The death was therefore consequent to a physical effort made during a very hot and humid day.

Temperatures that day were hotter than average. In fact, the average temperatures of the period (August 2020) in that time slot (2:00 pm–4:00 pm) are around 30 °C. In the present case, however, the temperature was 39.1 °C at 4:30 p.m. Furthermore, the average humidity of the period (August 2020) in that time slot (2:00 p.m.–4:00 p.m.) is 60%. Therefore, the degree of humidity was also above average.

## 3. Discussion

Global warming poses a substantial challenge to the health of the world’s population and contributes to the establishment of several diseases [7].

An interesting study published in the Lancet in 2015 evaluated a large sample (74 million deaths) on the incidence of exposure to suboptimal environmental temperatures (hot or cold) in the determinism of death of observations identifying a risk factor of 0.86% [16].

Although the relative risk of morbidity/mortality associated with extreme temperatures varies widely across studies, it is undisputed that there is a close correlation between rising environmental temperatures and cardiovascular disease [7,8].

Several studies have delved into this issue trying to identify the pathophysiological mechanism underlying this phenomenon.

One of the most accepted theories identifies an increase in skin blood flow and sweating in response to heat exposure, resulting in dehydration as a direct consequence of heat exposure. The hyper-viscosity condition resulting from hemoconcentration would be responsible for thromboembolism, representing a risk factor for ischemic events [17].

In fact, hyperthermia-altered vascular endothelium induces occlusion of arterioles and capillaries (microvascular thrombosis) or excessive bleeding (consumption coagulation), which could lead to multiorgan system failure, including cardiovascular dysfunction [16,17,18,19].

In addition to this phenomenon, there is the redistribution of blood flow in favor of the skin district aimed at dissipating internal temperature. This compensatory mechanism results in reduced intestinal blood flow, which, if protracted, is responsible for increased permeability of the intestinal epithelial membrane, allowing bacteria to pass through the intestinal lumen into systemic circulation [20].

Another theory supporting the influence of climate change on cardiovascular disease concerns the correlation with increases in CO_2_ and the consequent accumulation at ground level due to higher density than O_2_. The increase in the fraction of CO_2_ in the breathed air causes alterations at the level of the respiratory chain, resulting in the accumulation of protons and oxalate, known risk factors for heart disease [21,22].

Moreover, a 20-year longitudinal study conducted on a large sample showed a decline in renal function as environmental temperatures increased [23].

Animal studies have shown increased reactivity of the renin-angiotensin system followed by temperature-induced activation of the sympathetic nervous system [24].

In Italy in 2022, the Inail (The National Insurance Institute against Accidents at Work) released a guide with targeted recommendations for effective prevention of heat-related diseases in the workplace. In fact, the impact of extreme temperatures is particularly risky both for those who carry out their work in environments where it is not possible to achieve the conditions of comfort due to constraints linked to production needs or environmental conditions, and for those who work outdoors in the open, such as in agriculture and construction [25].

In the case examined, we deal with a man in previous apparent good health, who died suddenly at work. In fact, the environmental conditions detected together with the results of the medical–legal examinations carried out (diffuse congestion, absence of trauma and absence of gross acute pathologies capable of determining the rapid death) allowed us to identify the cause of death as a heat stroke.

In fact, in the workplace, heat exposure is an increasing challenge, especially for sensitive occupations that are constantly exposed to direct sunlight, first and foremost agricultural and construction workers.

Forensic diagnosis of causes of death uses standardized procedures. These procedures extend from inspection investigations to the execution of the autopsy, histological and laboratory investigations (including toxicological investigations) [26].

It is not always possible to draw certain diagnoses from these tests as numerous natural pathologies or external causes of various nature can determine pictures of little macroscopic or microscopic relevance.

In such situations, the forensic pathologist assumes that there may be a variety of etiologies leading up to death. Among these, the most common are malignant arrhythmias or acute dysmetabolic states.

It is, therefore, necessary to carry out an examination based on the victim’s previous clinical conditions (which can sometimes be completely unknown) and on historical-circumstantial data to ascertain whether documentable external factors (for example, physical effort or climatic factors) may have caused the death event.

This issue is correctly addressed both in the identification of the causal agent for compensation in the social security field (in Italy, INAIL) and in the field of employment insurance for public employees.

In the case report, the exact determination of the events that led to the subject’s death was very important precisely in the social security sphere.

Furthermore, the evaluation of the specific characteristics of the work carried out by the subject, correlated with the environmental conditions in which it was carried out, in the absence of specific anatomical–pathological findings of natural pathologies, allowed to identify the physical effort–climate interaction as a trigger of the acute event of which the subject was a victim. In other words, even referring to the classic concept of the id “quod plerumque accidit” (what is harmful to anyone), the identifiability of the event was evident, excluding those natural pathological conditions which, moreover, were not confirmed by the medico-legal examination.

In the present case, the subject was a farmer, and was frequently exposed to direct sunlight. In the circumstance, the environmental conditions of temperature and humidity were particularly unfavorable for the performance of a heavy manual activity (ambient temperature 39.1 °C with humidity 80%).

Solar radiation, humidity and wind are all factors favoring alterations related to heat exposure.

Some authors evaluated the correlation between heat exposure and 63,720 construction worker injury claims, where they found a 0.5 percent increase in the odds of traumatic injury per 1 °C increase in the daily Humidex index. In contrast, other studies have shown inverted U-shaped associations between heat exposure and occupational injuries.

However, these data need to be contextualized with the characteristics of the subject (ethnicity, gender, pathologies, therapy) and of the accident risk reduction measures taken by the company, such as stopping work shifts early on hot days [27].

Given the correlation between high environmental temperature and injuries, it is easy to understand how this phenomenon brings inevitable consequences in terms of economic and social costs [28].

In Italy, the employer is obliged to protect the health and physical and moral integrity of the employee (Article 2087 of the Civil Code).

Legislative Decree 81/2008 in Italy is a document with provisions on the protection of health and safety in the workplace. In fact, according to the Consolidated Occupational Health and Safety Act (Legislative Decree 81/2008), the employer must assess all risks arising from exposure to physical agents, including microclimate, making the most appropriate measures to protect the worker’s health. In this regard, social shock absorbers are provided to workers in cases of extreme weather events, including excessive temperature rise. In particular, companies whose workers have to deal with perceived temperatures of more than 35 degrees may apply for layoffs, taking into account the particular type of work.

Examples include road surfacing work, resurfacing work on building facades and roofs, outdoor work that requires protective clothing and also all work phases that, in general, take place in places that cannot be protected from the sun or that involve the use of materials or the performance of work that cannot withstand strong heat. These provisions are contained in the INPS (National Social Security Institute) circular no. 139/2016. In Italy, INPS is the main social security institution of the Italian public pension system.

However, the potential impacts of occupational heat exposure are underestimated due to under-reporting of heat illness and lack of awareness of heat-related injuries. Thus, on the one hand, workers need training to avoid heat illness and recognize symptoms in themselves and colleagues, and on the other hand, effective preventive measures are needed for occupational health and safety.

## 4. Conclusions

Workers, partly as a result of the increase in environmental temperatures in recent years, are exposed to a new “climatic” risk, related to an increase in illnesses associated with exposure to higher temperatures than a few decades ago. This is especially true for individuals employed in sensitive jobs working in climatic conditions that are more exposed to climatic variations [29].

The analysis provided in this study allows for some reflections on the need to adapt workplaces to the needs of individual workers, especially for tasks in which individuals are exposed to greater environmental risks, in accordance with current legislation.

Currently, the impact of these climatic variations on workers’ health is also underestimated in relation to the underreporting of heat-related illnesses.

A possible solution to this recently emerged problem, in the opinion of the authors, could be found by focusing on multidisciplinary adaptation strategies that incorporate climatology, indoor/building environments, energy use, work regulatory perfection and human thermal comfort models.

## Figures and Tables

**Figure 1 ijerph-20-04028-f001:**
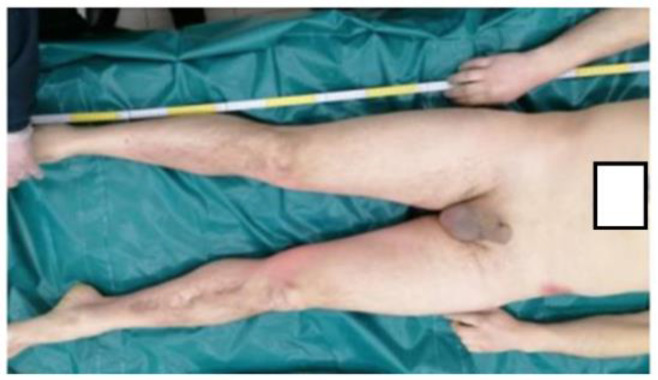
Anterior surface of the body.

**Figure 2 ijerph-20-04028-f002:**
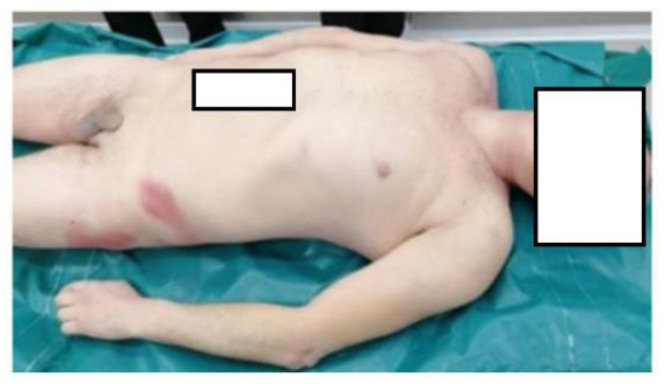
Details of previous image.

**Figure 3 ijerph-20-04028-f003:**
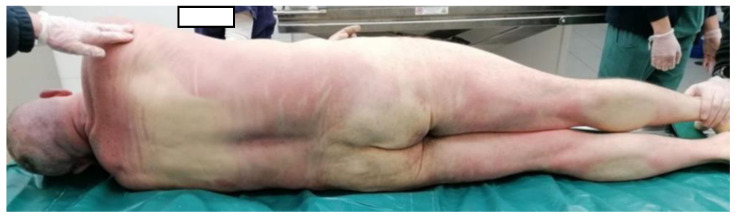
Posterior surface of the body.

**Figure 4 ijerph-20-04028-f004:**
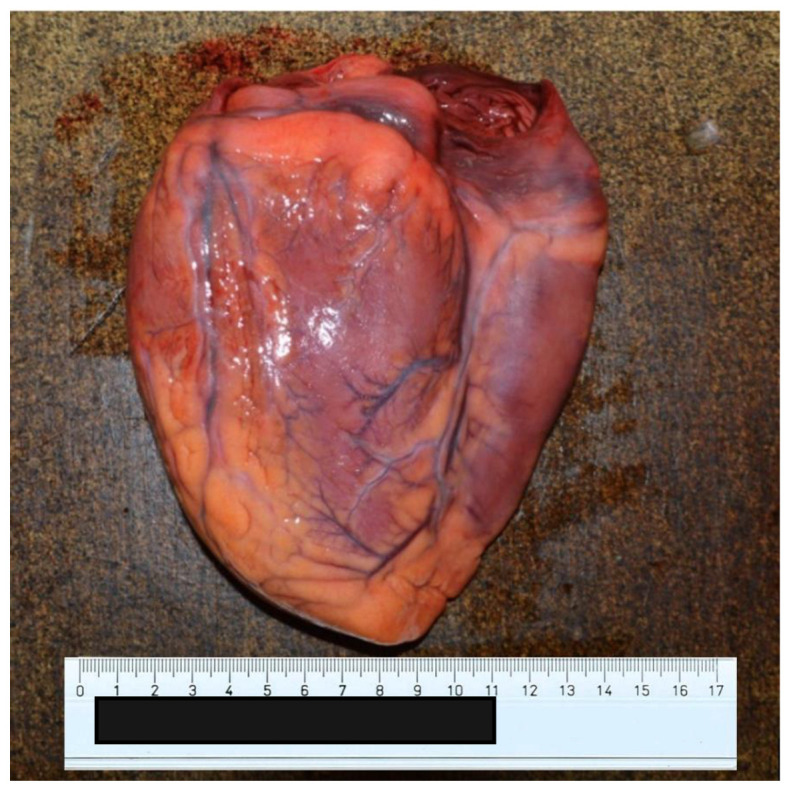
Anterior surface of the heart.

**Figure 5 ijerph-20-04028-f005:**
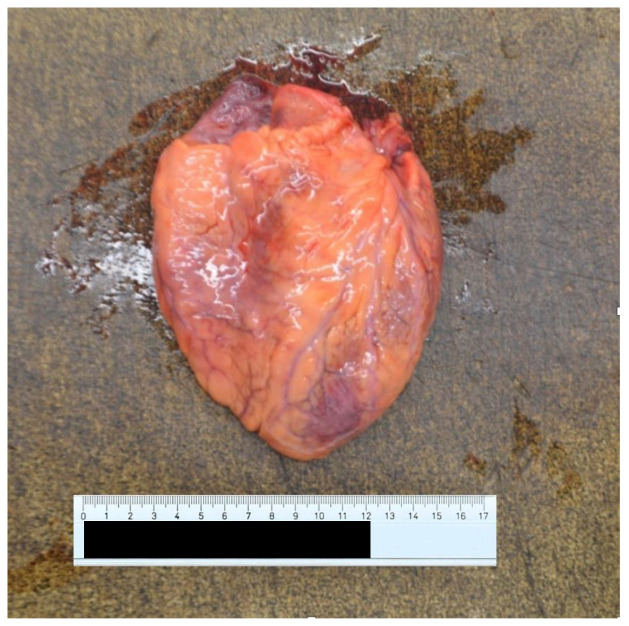
Posterior surface of the heart.

**Figure 6 ijerph-20-04028-f006:**
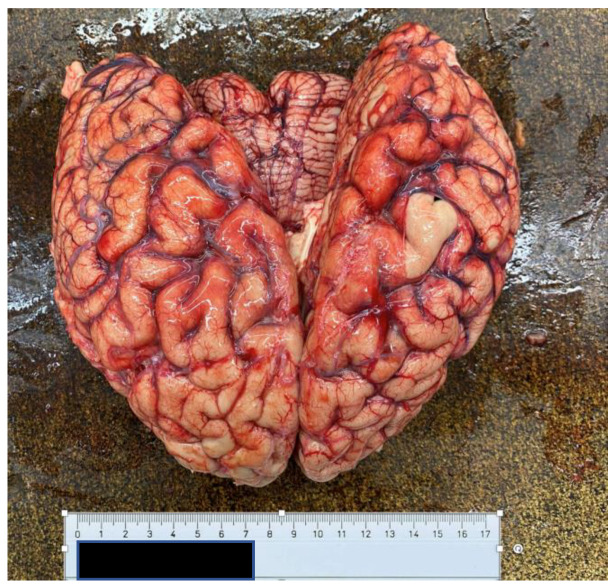
Anterior surface of the brain.

**Figure 7 ijerph-20-04028-f007:**
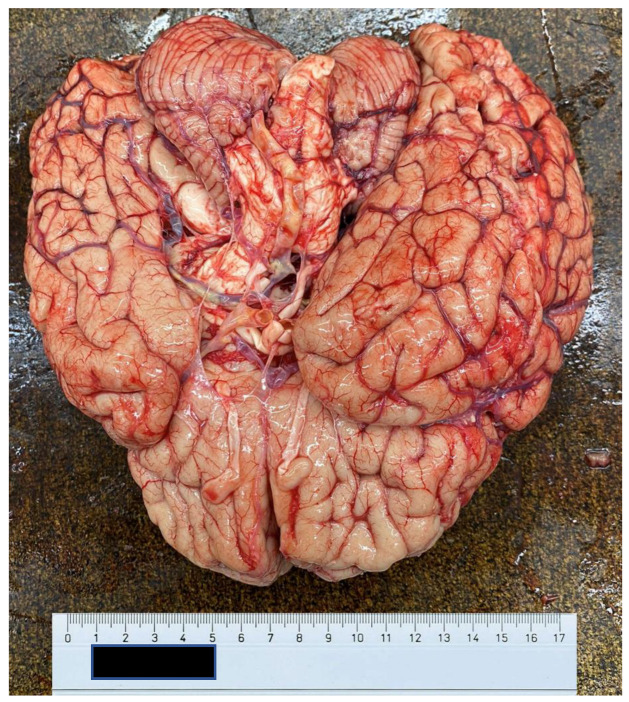
Posterior surface of the brain.

**Figure 8 ijerph-20-04028-f008:**
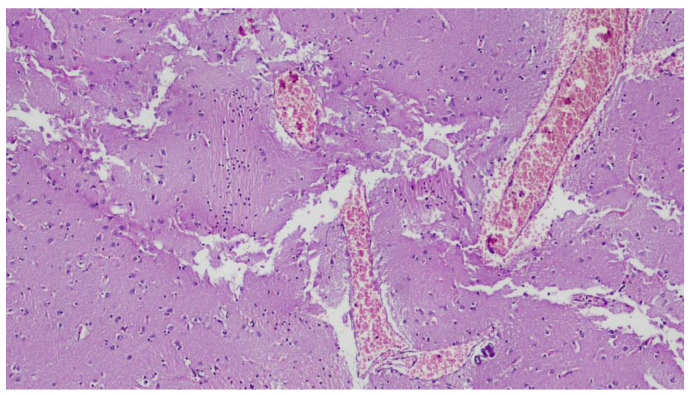
Histological micrograph showing brain cortex with ectasia of vessels and congestion (Hematoxylin-Eosin, Original Magnificatio 10×).

**Figure 9 ijerph-20-04028-f009:**
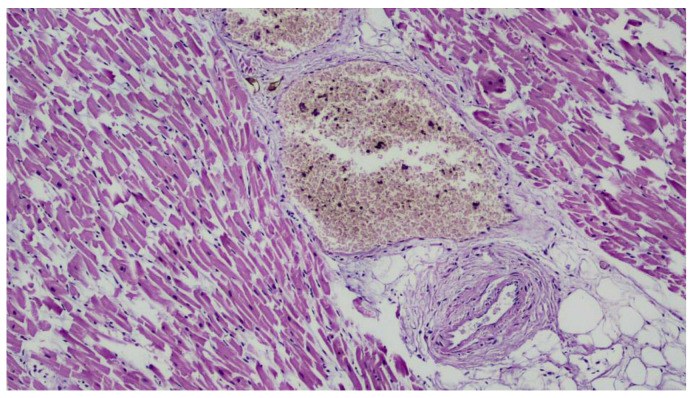
Histological micrograph showing congestion of vessels and fragmentation of muscle fibers (Hematoxylin-Eosin, Original Magnification 20×).

## Data Availability

Not applicable.

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
