# Peer review of "Heat Stroke in the Work Environment: Case Report of an Underestimated Phenomenon"

_ijerph, 2023, doi:10.3390/ijerph20054028_

Round 1

Reviewer 1 Report (Previous Reviewer 1)

Thank you for the opportunity of reviewing this revised manuscript. This paper presents the case report of the heatstroke during working. The manuscript was well revised. 

Author Response

Dear Reviewer n'1,

thank you very much.

Reviewer 2 Report (New Reviewer)

Dear authors, I read the proposed manuscript with great interest. 

I would like to congratulate your work. Global warming is now a reality and also represents a challenge for job protection. So it will be necessary to adapt the protocols and not just adopt the existing ones for higher temperatures. The article deals with a case of "death at work" caused by heat stroke.

However, there are a few aspects that the authors should clarify:

«The survey was conducted in the countryside of southern Italy. We arrived at the site about 90 minutes after the death (about 4:30 p.m.), and on the occasion, we found high ambient temperature (39.1 °C). High air humidity was also present. In fact, the hygrometer provided measured 80% humidity in the air. Ventilation was almost absent. The body was wearing a gray cotton T-shirt, a pair of dark blue jeans, and shoes of a dark-colored 89 leather-like material. Nothing significant was found at the external cadaveric inspection.» 

I think that PMI should be insert after the parameters of thanatochronology, unless there are witnesses. In this case I believe that the presence of the referred time must be inserted; then the parameters should be put in; in the end it must be deduced that the reported time and the normogram time are compatible. Please correct "Southern". 

Please add some photos: 

-external examination 

- macroscopic examination of Heart and CNS. 

- TC total body if performed to detect any form of violence

- histology of Heart [myocytolysis?], brain [swelling?]. 

- Do you performed IHC [HSP - Fineschi et al 2005?]

The autopsy report is ok. 

Illegal hiring is a problem in Italy. Workers - often immigrants - are forced to work long hours, even under the influence of amphetamines. Please, I think must be inserted toxicological analysis.

Please, I don't understand which country belong the law 81/2008, and explain what is "(INPS Circular 216 No. 139/2016)" at the international readers. 

Citation 30 AND citation 31 must be deleted. I read it, they have nothing to do with the text.

Author Response

Reviewer n'2

«The survey was conducted in the countryside of southern Italy. We arrived at the site about 90 minutes after the death (about 4:30 p.m.), and on the occasion, we found high ambient temperature (39.1 °C). High air humidity was also present. In fact, the hygrometer provided measured 80% humidity in the air. Ventilation was almost absent. The body was wearing a gray cotton T-shirt, a pair of dark blue jeans, and shoes of a dark-colored 89 leather-like material. Nothing significant was found at the external cadaveric inspection.» 

I think that PMI should be insert after the parameters of thanatochronology, unless there are witnesses. In this case I believe that the presence of the referred time must be inserted; then the parameters should be put in; in the end it must be deduced that the reported time and the normogram time are compatible. Please correct "Southern". 

Answer n'1: We added that the time of death was a reported datum and that the thanatochronological parameters detected did not conflict with it.

Reviewer n'2

Please add some photos: 

-external examination 

- macroscopic examination of Heart and CNS. 

- TC total body if performed to detect any form of violence

- histology of Heart [myocytolysis?], brain [swelling?]. 

- Do you performed IHC [HSP - Fineschi et al 2005?]

The autopsy report is ok. 

Illegal hiring is a problem in Italy. Workers - often immigrants - are forced to work long hours, even under the influence of amphetamines. Please, I think must be inserted toxicological analysis.

Please, I don't understand which country belong the law 81/2008, and explain what is "(INPS Circular 216 No. 139/2016)" at the international readers. 

Citation 30 AND citation 31 must be deleted. I read it, they have nothing to do with the text.3. 

Answer n'2: We have added photos of the external examination and of the macroscopic and microscopic features of the heart and brain. 3. CT was not performed.
4. IHC was not performed.
5. we have better specified the normative references, as requested.
6. we have eliminated bibliography 30 and 31.
Thanks again for the contribution.

This manuscript is a resubmission of an earlier submission. The following is a list of the peer review reports and author responses from that submission.

Round 1

Reviewer 1 Report

Thank you for the opportunity of reviewing this manuscript. This paper presents the case report of the heatstroke during working. The manuscript was well written except discussion section. Therefore, I believe some points must be modified to publish the article. 

Case presentation

l  The authors presented the case who died in a work environment. In this time, was the temperature and humidity higher than average temperature and humidity? Was the working place external or internal? Please write the detailed working environment.

Discussion

l  Discussion section must be rewritten. The construction is not good. For example, the one sentence forms one paragraph and what you would like to indicate spans multiple paragraph. The author should summarize what you would like to indicate in one paragraph.

l  In discussion, authors wrote the awareness of heat-related injuries (L. 163, 177). Is there a policy about the heat stroke prevention at industrial site in Italy? For example, in Japan, a policy about the heat stroke prevention at industrial site is performed by the Ministry of the Environment and the Ministry of Health, Labor and Welfare. If the prevention is poor in Italy, you may write some recommendations on prevention. 

Reviewer 2 Report

This work is mainly focusing on the health outcome in a work environment, so the authors should focus much more on the relationship between work environment and heart diseases, for the background and discussion. 

Climate change is related to the topic of this paper, but not most relevant. The authors can discuss how climate change and its resulted temperature increase impact human cardiovascular health. However, I think the main text should concentrate on the work environment, or more specifically, occupational health. There have been many studies looking into the adverse impact of climate change on cardiovascular diseases. If the authors focus on this aspect there is no novelty here. 

Reviewer 3 Report

I read your submission in fact your case report. i think it will improve if you explain a little bit more about case report concept, benefits, strength and definition.  furthermore, i suggest it is very useful for reader if authors can add some line about methodology. 

also i suggest authros read this article:

Understanding the influence of Iranian farmers’ climate change beliefs on their adaptation strategies and mitigation intentions